# Differences in Growth–Climate Relationships among Scots Pines Growing on Various Dune Generations on the Southern Baltic Coast

Anna Cedro [1],*, Bernard Cedro [1] and Marek Podlasiński [2]

[1] Institute of Marine & Environmental Sciences, University of Szczecin, Adama Mickiewicza 16, 70-383 Szczecin, Poland; bernard.cedro@usz.edu.pl

[2] Department of Environmental Protection, West Pomeranian University of Technology in Szczecin, Słowackiego 17, 71-434 Szczecin, Poland; marek.podlasinski@zut.edu.pl

\* Correspondence: anna.cedro@usz.edu.pl

**Abstract:** This study focuses on analyzing the growth rate and growth–climate relationship in Scots pine (*Pinus sylvestris* L.) growing on coastal dunes of different ages on the Southern Baltic coast. Podzols have developed on these dunes. Depending on dune age, however, podzols are characterized by a different degree of development and richness, which represents the main factor differentiating the studied habitats: the oldest brown dunes (BD), younger yellow dunes (YD), and the youngest white dunes (WD). Samples were taken from 68 trees using Pressler borers. Using classic cross-dating methods, local chronologies were compiled that served as the basis for further analyses. Basic analyses of soil properties were also performed. Trees growing on brown dunes display the widest tree-rings and the highest rate of cumulative radial growth. Both rates are the lowest for trees growing on white dunes (WD). The dominant meteorological factor shaping tree-ring widths is late winter/early spring air temperature (February/March). However, in poorer habitats with inferior soil air–water conditions, rainfall sums and rainfall distribution through the year become progressively more significant factors. On white dunes (WD), the strongest growth–climate correlations are obtained for summer precipitation. These results identify habitat richness as the main factor shaping growth dynamics in Scots pines growing on dunes. Due to the protective function of the studied tree stands (coast protection), and in light of the rising sea levels and increasing storm intensities, further studies are required, aiming at understanding all interrelationships occurring in these valuable ecosystems.

**Keywords:** dunes; podzols; Scots pine; *Pinus sylvestris* L.; tree-ring width; dendroclimatology; Baltic Sea; Poland

## 1. Introduction

Scots pine (*Pinus sylvestris* L.) is one of the most common subjects of dendrochronological analyses, i.a., due to its broad distribution in both Europe and Asia, as well as mass plantings in numerous countries [1–3]. The growth–climate relationships for this species are well-known from various locations (distinct climate zones), various substrates (e.g., fens, dunes), and from areas prone to various impacts, both natural (e.g., insect gradations or wildfires) and anthropogenic (e.g., air, soil or groundwater pollution) [4–15]. Pine-based chronologies spanning multiple centuries are the basis for dating archeological objects, works of art, various types of wooden constructions, and utensils [9,14,16–19]. In the Southern and Eastern Baltic coast regions, Scots pine is also the main focus of dendrochronological studies, e.g., [12,13,20–23], apart from the native oak species (*Quercus robur* and *Q. petrea*). Numerous studies point to late winter and early spring air temperatures (February and March) as the dominant factor shaping tree-ring width in Scots pine [4,8,9,11,24–26]. Climate warming observed in recent decades (especially affecting the winter–spring transition) causes the vegetation season to become extended, and results in improved pine growth

conditions in this region [25,27]. Additionally, the atmospheric precipitation sums and rainfall distribution through the year influence the growth dynamics in this species. A positive impact of rainfall in the spring and summer periods on tree-ring width is observed especially at dry sites (groundwater residing at large depths, sandy substrate) [21,25,26].

Through the past several thousand years, the chosen study area underwent a profound transformation, often displaying a catastrophic character. Due to the rapid retreat of the most recent ice sheet, the study area was covered by fluvioglacial sediments and landforms, and the coast of the nascent Baltic Sea was located over 50 km north relative to the present-day coastline. About 7300 years ago, as a result of progressive Baltic Sea level rise, and a series of storms, the coastline was disrupted, and a gulf was developed, reaching several tens of kilometers to the south [28]. Islands formed by morainic plateaus of Uznam and Wolin were intensely abraded by waves and sea currents, which contributed to the formation of the Świna Gate, along with its diachronous series of dune ridges [29,30]. These dunes, inhabited by Scots pine, are a remarkably interesting site, enabling observation of podzolization processes taking place in the same climatic conditions, on the same substrate, with the same vegetation, but differing in the duration of pedogenetic processes. At the same time, the dunes, and the tree stands growing on the dunes provide direct coast protection against rising sea levels, storm surges, and high waves caused by air pressure drops and winds associated with low-pressure systems that in recent years are becoming progressively deeper and are gaining intensity.

This study aims to: (1) analyze the rate of tree-ring growth, (2) determine the influence of the soil-forming process development and soil richness on tree-ring width in Scots pine, and (3) examine the growth–climate relationship in trees growing on the Świna Gate dune series of different ages.

## 2. Material and Methods

### 2.1. Study Area

The study area is located in the Świna mouth area, on the southern coast of the Pomeranian Bay (Figure 1). Świna is one of three outlets of the Odra River to the Baltic Sea, from east to west: Dziwna and Świna, with outlets in Poland, and Peene, which has an outlet in Germany. The Świna Gate is comprised of forms that arose due to marine, fluvial and aeolian accumulation. Groups of ridges of aeolian landforms that developed here were first described by Deecke [31] and Keilhack [32,33]. Keilhack distinguished three generations of dunes that, in his opinion, corresponded to three stages of the Świna Gate formation: brown dunes (BD), yellow dunes (YD), and white dunes (WD). Geomorphological diversity and pedogenetic maturity were the basis for these distinctions. Subsequent studies by Osadczuk [34] distinguished a further generation of white dunes (WD-I, white I). Dune processes taking place in the discussed area were studied also by Reimann et al. [35], Łabuz [36], Dudzińska-Nowak [37], and Lampe and Lampe [30]. The latter study distinguished seven dune generations on part of Wolin Island (E1 through E3—brown dunes, E4 and E5—yellow dunes, and E6 and E7—white dunes), and two generations on part of Uznam Island (W1-W2) (Figure 1). According to Lampe and Lampe [30], the OSL age of brown dunes (E1–E3) ranges from $2.45 \pm 0.15$ to $5.39 \pm 0.37$ ka. The yellow dunes (E4 and E5) OSL age ranges from $0.59 \pm 0.03$ to $1.72 \pm 0.12$ ka. The white dunes (E6 and E7) OSL age ranges from the recent to $0.45 \pm 0.02$ ka.

Dunes of all generations are composed of well-sorted quartz sands very poor in mineral elements. The humid climate favors the leaching of nutrients from the soil profile, and sands have low pH buffering capacity, which controls the development of the podzolization process [38]. Although initially not evident (as in the case of WD), with time this process leads to the development of podzols (YD and BD). Such soil chronosequence occurs in many places on Earth [39–41]. As a result of the soil-forming process lasting about 400 years, a poorly developed podzol occurs on the youngest white dunes (WD, E6 dune series), characterized by the following soil profile: O-AE-BC-C (Figures 1 and 2). Podzols developed on the 1600–1700 years old yellow dunes (YD, E4 dune series) and on the oldest

brown dunes (about 2500 years old, series E3). These podzols display the following soil profiles: O-A-Es-Bs-BC-C and O-A-Es-Bhs-Bs-C, respectively (Figures 1 and 2). Uneven soil development on dunes of differing ages is also reflected in the chemical properties of individual genetic horizons.

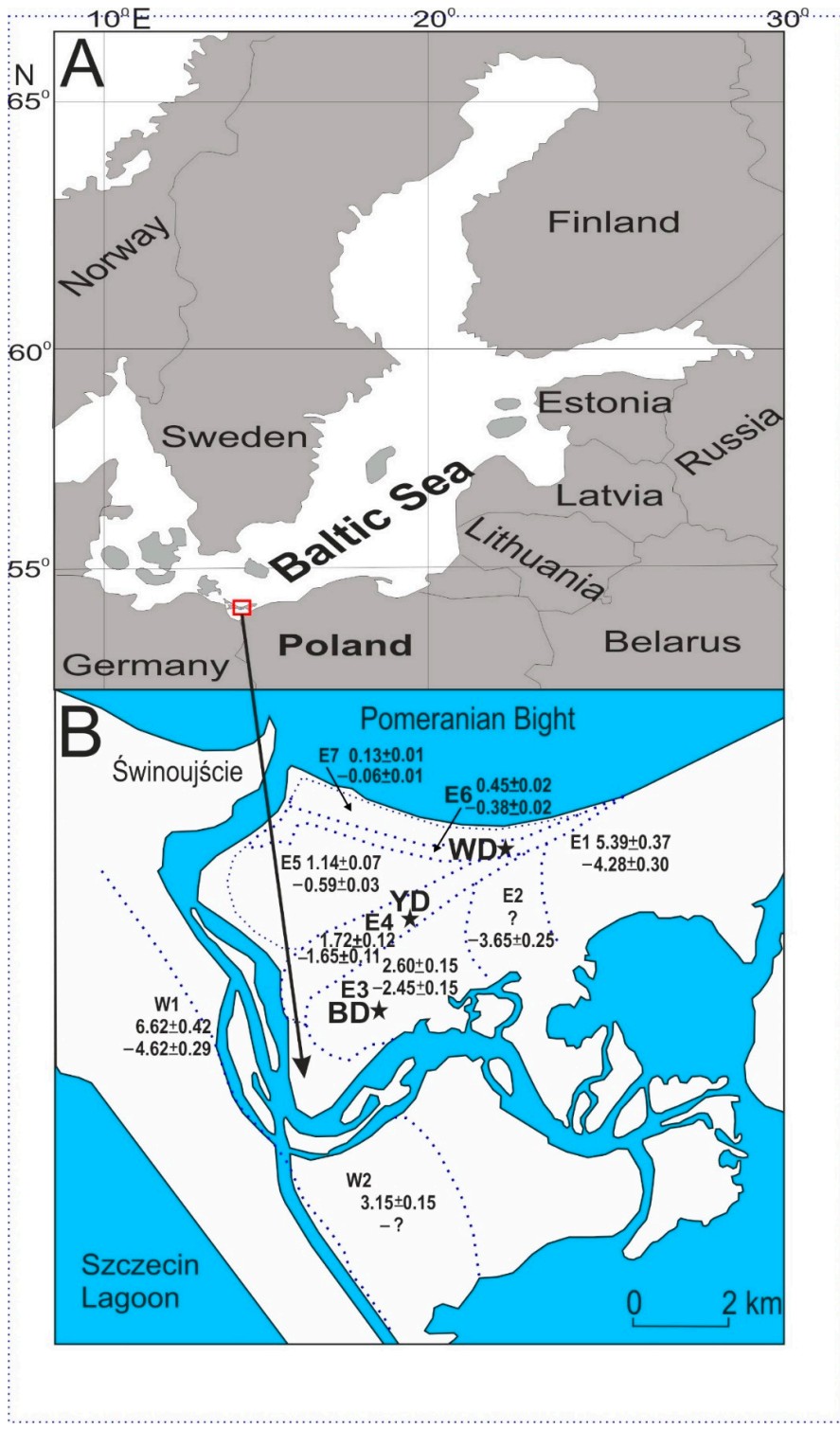

**Figure 1.** Study area location: (**A**)—relative to the Baltic Sea basin; (**B**)—relative to the Świna Gate. E1–E7 and W1–W2: dune generations and OSL datums following Lampe and Lampe [30], stars—location of study surfaces: BD—brown dunes, YD—yellow dunes, WD—white dunes.

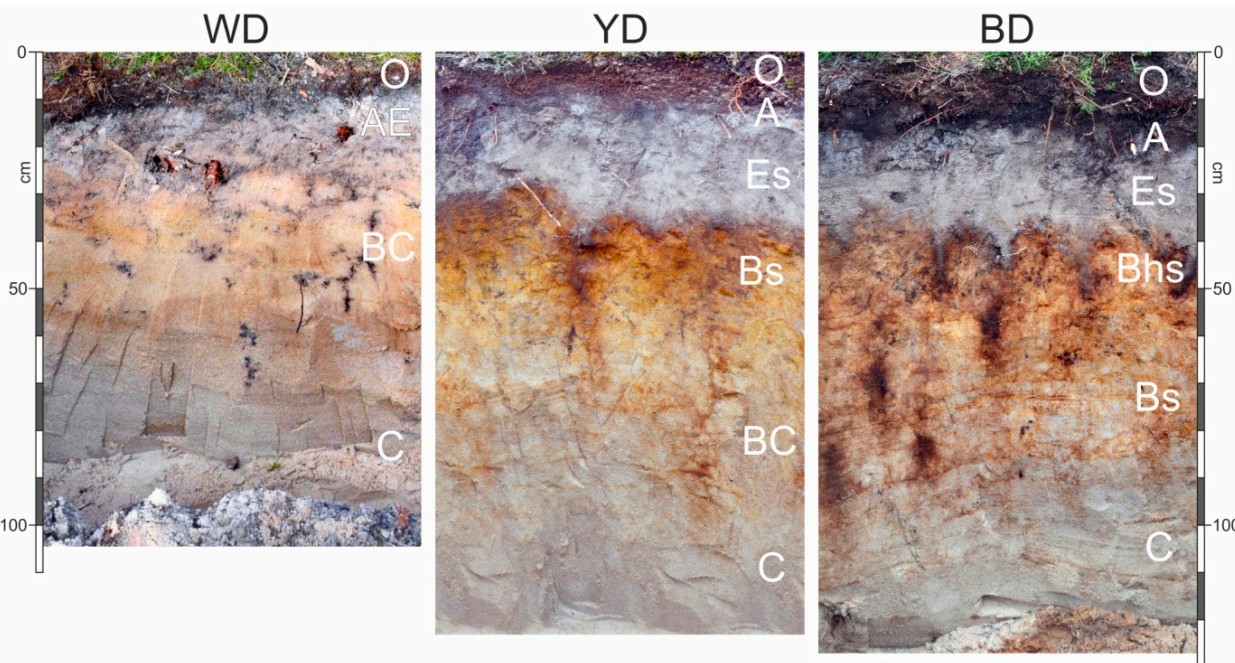

**Figure 2.** Soil profiles for white (WD), yellow (WD), and brown dunes (BD). Soil genetic horizons: O—organic horizon, A—humus horizon, E—eluvial horizon, B—illuvial horizon, C—parent material, s—leaching of (or enrichment in) iron and aluminum, h—illuvial accumulation of organic matter.

The Świna Gate dunes are vegetated by pine forests of variable ages, with a minor contribution of other taxa (e.g., spruce or oak). Due to the protective functions of these forests (protection of coast and dunes), large areas are occupied by forests planted in the 19th century and in the first decades of the 20th century.

*2.2. Tree-Ring Data*

Samples were taken in late August 2018 from trees dominant or co-dominant in a stand, growing on the summits of low dunes, using a Pressler borer, at 1.3 m height above ground. A total of 68 trees were sampled, obtaining 74 measuring radii, with one sample per tree. An additional sample was collected from a given tree in case of sample destruction or when any anatomic abnormalities were observed. In the laboratory, samples were glued onto boards, dried, and sliced with a knife in order to obtain a clear view of the tree-rings. In order to enhance tree-ring boundaries, the sample surfaces were smeared with chalk. Tree-ring width was measured under a stereoscopic microscope down to 0.01 mm, using DENDROMETER 1.0 software [42]. A total of 7371 tree-rings were measured, including 10 frost rings (deformations associated with extreme weather conditions—drop of temperature below 0 °C during the growing period, which results in tissue dehydration and deformation of vessels, tracheids, and cells). Local chronologies were subsequently compiled using classic cross-dating methods. Based on the high visual similarity among dendrochronological curves, and high values of statistical indices (Student's t-test and correlation coefficient), dendrochronological sequences were selected for inclusion in the chronology. Thirteen sequences that were the least visually and statistically correlated were rejected. Chronology robustness was tested using COFECHA, part of the DPL software package [43–45]. Student's t-test and coherence coefficient (Gleichläufigkeitswert, Gl) were computed for pairs of chronologies using the TCS 1.0 program [46], in order to determine the similarity between local chronologies. The EPS coefficient was also computed [47]. Age trend and autocorrelation were subsequently removed from the dendrochronological sequences selected for the chronology by means of an indexing process (a two-phase detrending technique, by fitting either a modified negative exponential curve or a regression line with a negative or zero slope) [43,44,48,49].

Monthly mean air temperatures and monthly precipitation totals spanning June of the year preceding growth (pVI) through September of the vegetation year (IX) were used for correlation and response function analysis. The analysis was performed separately for temperature and precipitation, which yielded $r^2$ values (multiple regression determination coefficients) for each meteorological parameter [50–52]. Pointer year analysis was carried out using TCS software [46] by calculating positive years (+) characterized by an increase in tree-ring width relative to the preceding year, and negative years (−), with a reduction in tree-ring width relative to the preceding year [53,54]. Pointer years were calculated based on a minimum of 10 trees, assuming 90% as the minimum incremental trend consistency threshold. In order to facilitate pointer year comparison among all 3 chronologies, pointer years were presented as dimensionless values, following the formula:

$$\frac{\text{growth per year in a given pointer year}}{\text{multi-year mean growth per year}}$$

Thus, positive values were higher than multi-year mean growth per year, and negative values were lower than multi-year mean growth per year [55].

### 2.3. Sand and Soil Analysis

Soil profiles were examined in March 2021 within the same forest surfaces that were sampled for dendrochronological analyses.

Grain size analysis was performed for 13 sediment samples collected from three dune generations occurring in the Świna Gate. Dried samples weighing 100 g were sieved on a sieve shaker through a standard set of sieves manufactured by Fritsch. Using GRADISTAT MACRO (version 5_11_PL) for Microsoft Excel and the formulae of Folk and Ward [56], graphical grain size indices were determined: mean diameter (Mz), standard deviation ($\sigma_I$), skewness ($Sk_I$), and kurtosis (KG). Sorting classes, skewness types, and peakedness of grain size distribution curves were assumed in accordance with the ranges defined by Folk and Ward [56].

Organic matter content (%) was measured by loss on ignition (LOI) at 550 °C. Soil pH was determined potentiometrically in 1 M KCl solution. Organic carbon (Corg) content (%) and total nitrogen (Ntot) content (%) were determined using a CNS Thermo Fisher Scientific elemental analyzer. Fe content was determined using ICE spectrometry on a Solar 3000 spectrophotometer, after mineralization of a one-gram sample in a 1:1 mixture of concentrated nitric and perchloric acids.

### 2.4. Climate Data

Data from the nearby weather station at Świnoujście (IMGW network station no. 122000, 53.91° N 14.25° E, 6 m a.s.l.) were utilized for dendroclimatological analyses (correlation and regression analysis, and pointer year analysis). The station is located 7–11 km to the west of the study area. Meteorological data (air temperature and rainfall, monthly values) spanned the period 1948–2018 (71 years). The mean annual air temperature for this period for Świnoujście equals 8.6 °C. The warmest year was 2014 with T = 10.0 °C, and the coldest year was 1996 with T = 6.9 °C. The coldest month was January, with a mean temperature equal to 0.0 °C (ranging from −7.0 to 5.1 °C), although the lowest monthly mean temperature recorded at this station was −7.5 °C in February 1956 (Figure 3). The warmest months are July (17.6 °C, ranging from 14.5 to 21.8 °C) and August (17.5 °C). Mean annual precipitation sum equals 564 mm. The driest year was 1982 with an annual precipitation total equal to 377 mm. The most humid year was 2017, with a rainfall sum equal to 783 mm. The highest monthly rainfall sums are noted for July: 64 mm (ranging from 0 to 255 mm), June: 58 mm (ranging from 8 to 146 mm), and in August: 57 mm (ranging from 7 to 185 mm) (Figure 3). The lowest precipitation sum is observed for February (31 mm). In the Köppen–Geiger classification, the climate of this area is classified as Dfb (cold climate, without a dry season and warm summer) [57].

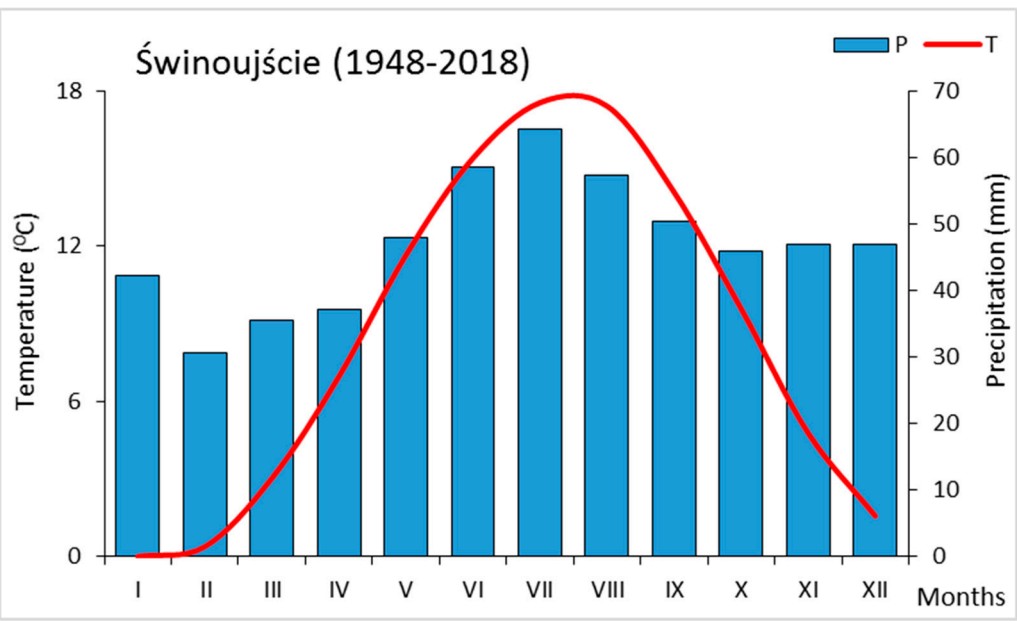

**Figure 3.** Mean monthly temperature (T) and precipitation (P) values in Świnoujście through the period 1948–2018 (71 years).

## 3. Results

### 3.1. Aeolian Sands and Soils

Statistical parameters of the grain size distribution for the analyzed sands computed following the graphical method according to the formulae of Folk and Ward [56] indicate no significant differences among the sands making up the BD, YD, and WD dunes. This is typical for aeolian sands [29]. The narrow scatter of Mz values (from 2.001 to 2.124) and $\sigma_I$ values (from 0.413 to 0.490) indicates stable dynamic conditions of the sedimentary environment and uniformity of the transport and deposition processes (Table 1). At the same time, the $Sk_I$ index (ranging from 0.010 to −0.307) points to variable transport conditions and airstream velocity, which resulted in the deflation of finer particles. Additionally, the kurtosis index KG (ranging from 0.973 to 1.150) points to variable energy conditions of the depositional environment.

**Table 1.** Mean grain size parameter values for aeolian sands. BD—brown dunes, YD—yellow dunes, WD—white dunes. Soil genetic horizons: O—organic horizon, A—humus horizon, E—eluvial horizon, B—illuvial horizon, C—parent material, s—leaching of (or enrichment in) iron and aluminum, h—illuvial accumulation of organic matter. Mz—mean diameter, $\sigma_I$—standard deviation, $Sk_I$—skewness, KG—kurtosis.

| Samples | $M_z$ | $\sigma_I$ | $Sk_I$ | $K_G$ |
|---------|-------|------------|--------|-------|
| BD/Es | 2.006 | 0.490 | −0.193 | 1.051 |
| BD/Bhs | 2.001 | 0.440 | −0.260 | 1.013 |
| BD/Bs | 2.009 | 0.459 | −0.246 | 1.013 |
| BD/C | 2.010 | 0.413 | −0.307 | 0.973 |
| YD/A | 2.027 | 0.495 | 0.010 | 1.067 |
| YD/Es | 2.044 | 0.470 | −0.044 | 1.044 |
| YD/Bs | 2.042 | 0.422 | 0.000 | 0.998 |
| YD/BC | 2.009 | 0.413 | −0.013 | 0.979 |
| YD/C | 2.124 | 0.413 | 0.101 | 1.150 |
| WD/AE | 2.016 | 0.460 | −0.280 | 1.091 |
| WD/BC | 2.009 | 0.418 | −0.321 | 0.978 |
| WD/C1 | 2.022 | 0.410 | −0.292 | 1.013 |
| WD/C2 | 2.022 | 0.407 | −0.295 | 0.988 |

Soils from the three studied sites represent podzols at different development stages. This is evident from the colors of the genetic horizons (Figure 2), and from some chemical properties (Table 2). The soil on the white dunes (WD) represents a poorly developed podzol, due to a distinctive whitening (AE horizon) about 12 cm below the organic horizon (O). Below this bleached horizon, there is a poorly developed illuvial horizon (BC), which transitions into the parent material (C). Due to the young age of white dunes, the leaching of humus and iron is not yet high, as indicated by the Fe content, which is higher than in the soils developed on older dune generations (Table 2). On yellow dunes (YD), below a 10 cm thick organic horizon, and a 20 cm thick humus horizon (A), there is a well-developed 10 cm thick light gray eluvial horizon (Es). Below, there is a yellow-brown illuvial horizon (Bhs), also 10 cm thick, grading (BC) into horizon C. Brown dunes (BD) are characterized by a thicker eluvial horizon (Es—15.3), of white-gray color, and a bipartite dark brown illuvial horizon Bhs and Bs, with a total thickness of 45 cm. The longer period of soil development on YD and BD dunes is also reflected in pH. In the studied soils, all surface horizons (O and A) are strongly acidic, but the pH value decreases with the increasing duration of the pedogenetic process (Figure 4).

**Table 2.** Selected chemical properties of soils from the study sites. BD—brown dunes, YD—yellow dunes, WD—white dunes.

| Dune Type | Depth (cm) | Symbol | pH H$_2$O | pH KCl | C Org. (%) | N Tot. (%) | C:N | Fe Tot. (mg·kg$^{-1}$) |
|---|---|---|---|---|---|---|---|---|
| BD | 0–15 | O | 3.2 | 2.5 | 30.9 | 1.0 | 29.4 | 3078 |
| | 15–25 | A | 3.4 | 2.6 | 8.2 | 0.3 | 30.3 | 1110 |
| | 25–45 | Es | 3.6 | 3.4 | 0.4 | 0.3 | 1.3 | 782 |
| | 45–60 | Bhs | 4.3 | 3.9 | 0.6 | 0.5 | 1.3 | 1825 |
| | 60–90 | Bs | - | - | - | - | - | - |
| | 90–120 | C | - | - | - | - | - | - |
| YD | 0–10 | O | 3.5 | 2.5 | 40.8 | 1.6 | 25.8 | 3351 |
| | 10–30 | A | 3.4 | 2.9 | 1.7 | 0.1 | 19.1 | 1159 |
| | 30–40 | Es | 4.1 | 3.4 | 0.4 | 0.1 | 3.7 | 839 |
| | 40–50 | Bs | 3.8 | 3.4 | 0.8 | 0.1 | 6.6 | 2222 |
| | 50–80 | BC | - | - | - | - | - | - |
| | 80–120 | C | - | - | - | - | - | - |
| WD | 0–12 | O | 3.4 | 2.5 | 44.8 | 1.4 | 31.6 | 1317 |
| | 12–25 | AE | 3.9 | 2.3 | 0.5 | 0.2 | 2.3 | 1234 |
| | 25–60 | BC | - | - | - | - | - | - |
| | 60–100 | C | - | - | - | - | - | - |

"-"—not determined, C org.—organic carbon, N—total nitrogen, Fe—total iron; for symbol explanations see Figure 2.

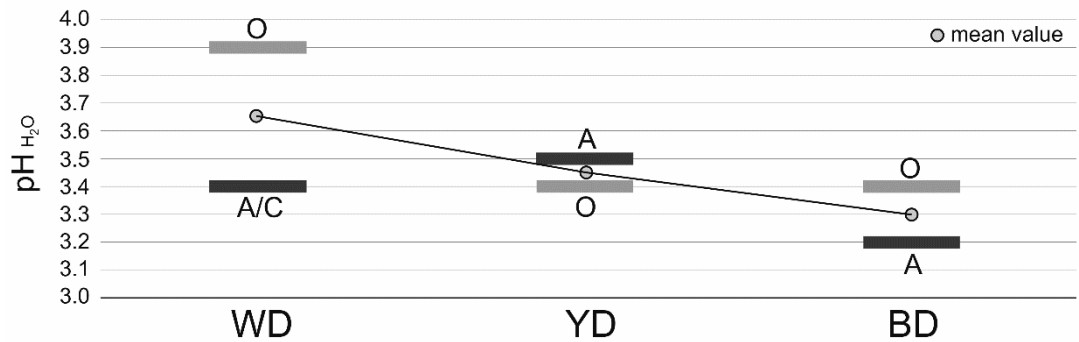

**Figure 4.** Reaction (expressed as pH) of organic (O) and humus (A) horizons in the studied soils. WD—white dunes, YD—yellow dunes, BD—brown dunes.

A distinctive feature of podzols is poor decomposition of organic matter and its accumulation in the form of an organic horizon (O). The carbon content in horizon O

decreases with increasing soil age (Table 3). This is likely due to smaller organic matter decomposition in younger soils (WD). However, the total quantity of organic matter accumulated in surface horizons (O + A) is the highest in the podzol developed on the brown dunes, and the lowest in the youngest soil, developed on the white dunes (Figure 5).

**Table 3.** Basic statistics of measured and index (residual) Scots pine local chronologies. Abbreviations: BD—brown dunes, YD—yellow dunes, WD—white dunes, TRW—tree-ring width, SD—standard deviation, 1AC—first-order autocorrelation, MS—mean sensitivity, EPS—expressed population signal.

| Lab. Code | No. of Years | Time Span | No. of Samples | Mean TRW (Min–Max) (mm) | Measured Chronology | | | Residual Chronology | | | EPS |
|---|---|---|---|---|---|---|---|---|---|---|---|
| | | | | | SD | 1AC | MS | SD | 1AC | MS | |
| BD | 93 | 1926–2018 | 21 | 1.93 (1.43–2.42) | 0.867 | 0.633 | 0.279 | 0.210 | −0.088 | 0.259 | 0.95 |
| YD | 92 | 1927–2018 | 22 | 1.83 (1.25–2.42) | 0.806 | 0.609 | 0.300 | 0.237 | −0.082 | 0.296 | 0.93 |
| WD | 139 | 1880–2018 | 18 | 1.03 (0.80–1.80) | 0.538 | 0.629 | 0.297 | 0.216 | 0.017 | 0.265 | 0.88 |

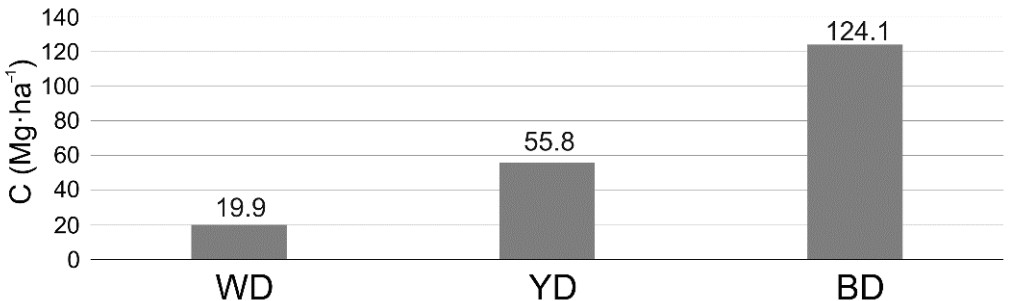

**Figure 5.** Total C stock in the soils developed on the studied dunes. WD—white dunes, YD—yellow dunes, BD—brown dunes.

*3.2. Ring-Width Chronologies*

The longest chronology was compiled for white dunes (WD). It spans 139 years from 1880 to 2018 (Figure 6, Table 3). At this site, however, trees are about 160 years old and were planted in the early 1860s (the oldest dated chest-height tree-ring adjacent to the core comes from 1867). Trees growing on yellow dunes (YD) and brown dunes (BD) are of similar age and were planted in the mid-1910s. The oldest dated chest-height tree-rings adjacent to the cores at these sites come from 1921 and 1920, respectively. Additionally, the chronologies have a similar temporal span: YD chronology is 92 years long (1927–2018) and BD chronology is 93 years long (1926–2018). The highest tree-ring width was measured in trees growing on brown dunes (1.93 mm). The lowest tree-ring width was observed in pines growing on white dunes (1.03 mm) (Table 3). The chronologies are highly coherent (Figure 6), both visually and statistically: the Student's t-test value for pairs of chronologies ranges from 9.17 to 12.06, and the Gl coherence coefficient ranges from 77 to 90%. A comparison of cumulative growth of the studied trees through the first 100 years of their lives indicates the highest radial growth rate for the trees growing on brown dunes, slightly lower for trees growing on yellow dunes, and considerably lower tree-ring widths (almost two times lower) in pines growing on white dunes (Figure 7).

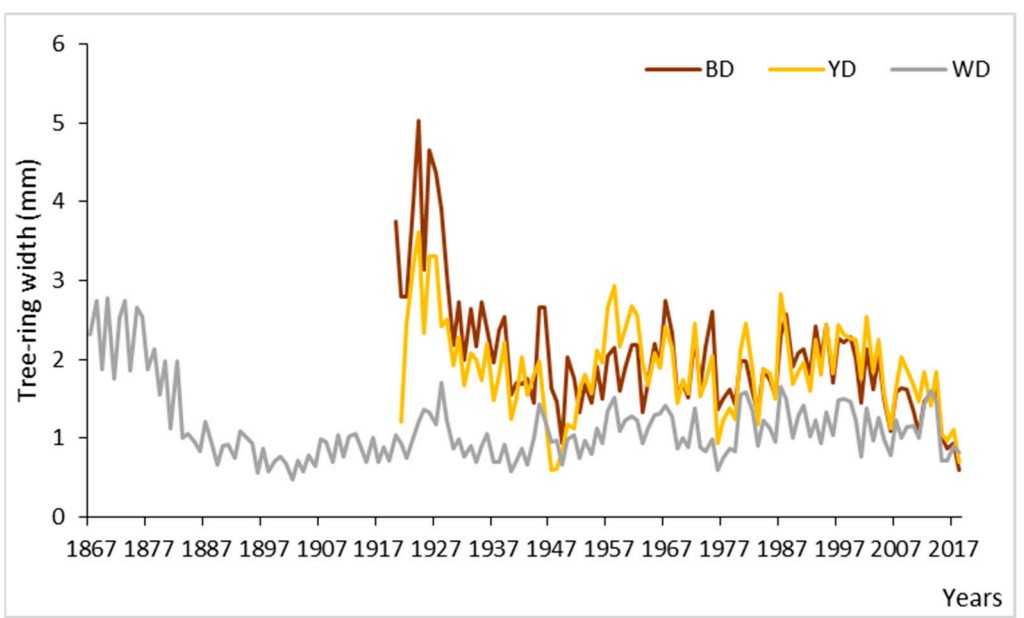

**Figure 6.** Local Scots pine chronologies (BD—brown dunes, YD—yellow dunes, WD—white dunes).

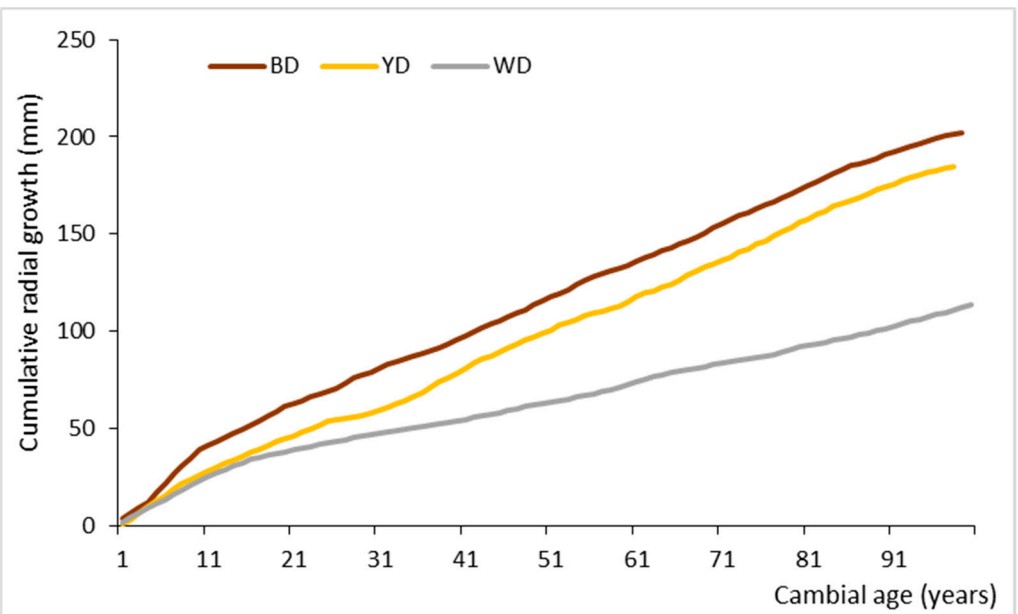

**Figure 7.** Cumulative radial growth of Scots pine from brown dunes (BD), yellow dunes (YD), and white dunes (WD) expressed in mm.

### 3.3. Dendroclimatological Analysis

3.3.1. Pointer Years Analysis

Pointer years were calculated for each population. A total of 109 (50+, positive, and 59–, negative) pointer years were obtained. For 21 years, in all the studied populations, the same growth reactions occurred in >90% of trees. Of these, in 9 years an increase in tree-ring width relative to the preceding year was noted (positive years: 1955, 1972, 1980, 1987, 1995, 1997, 2002, 2007 and 2012), and in 12 years, a decrease in tree-ring width relative to the preceding year was noted (negative years: 1940, 1947, 1959, 1963, 1969, 1973, 1976, 1989, 1996, 2003, 2006 and 2015) (Figure 8). Figure 8 shows indexed tree-ring widths against multi-year mean values during negative (negative values) and positive (positive values) pointer years. Three years stand out among these data: 1987, with the largest tree-ring

width increase (with a maximum for yellow dunes), and 1947 and 1976, with the lowest tree-ring width increase (again with a minimum for yellow dunes).

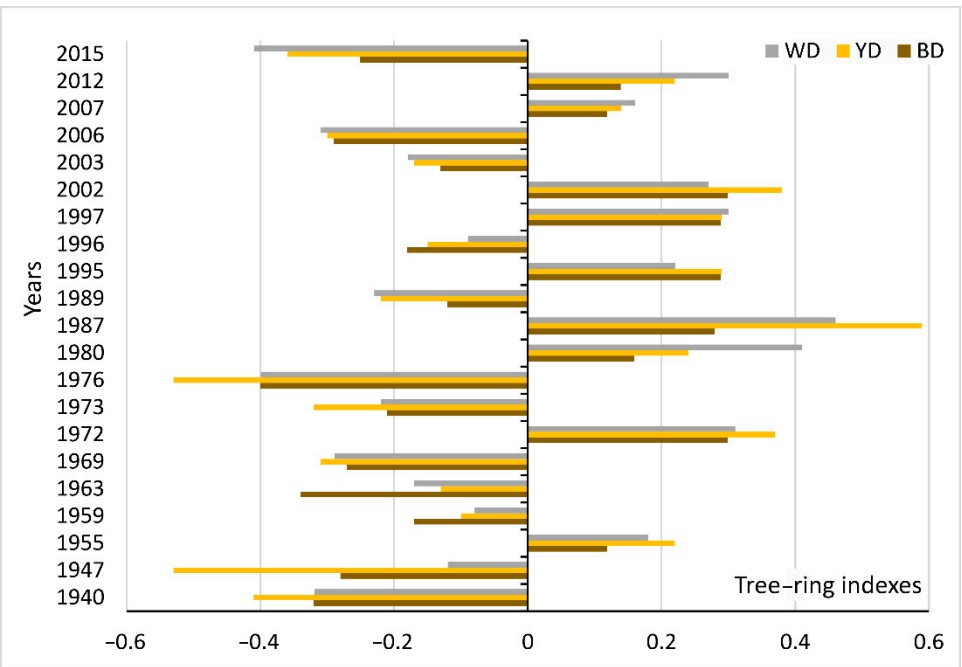

**Figure 8.** Tree-ring widths (indexes) in pointer years (converted to dimensionless values).

An analysis of weather condition variability in these years shows that a positive pointer year occurs in a vegetation season preceded by a warm and short winter, early and warm spring, and high precipitation from May to August. Negative pointer years occur in years with cold and long winters, often with negative air temperature occurring as late as March, with a delayed onset of spring, and shortages of rainfall in summer.

The analysis also indicated pointer years occurring in only one of the analyzed chronologies: for WD—only the year 1970 (positive), for YD—three years, including two positive (1965 and 2014), and one negative (2013), and for BD—seven years, including five negative (1954, 1956, 1966, 1994 and 2011), and two positive (1938 and 1974). The year 1970, characterized by an increase in tree-ring width in most trees from the WD population, was a very humid year, with annual precipitation sum 200 mm higher than the multi-year average. No dominant climatic factor determining the growth dynamics could be identified for pointer years observed only in the YD chronology (for instance, in the negative year 2013, March was very cold and frosty, while in positive years winter and early spring temperatures are variable, and precipitation remains on an average level). In the BD chronology, negative pointer years are characterized by a cold and frosty February, with March temperatures below the multi-year average. Precipitation shortages in summer (e.g., no rainfall in August 1994) are an additional factor. In the positive pointer year in the BD chronology (1974; no data for 1938), an increase in tree-ring width is noted, despite a precipitation deficit in the entire year, including summer months. This may be linked with high temperatures in the winter months and in March.

### 3.3.2. Correlation and Response Function Analysis

Analysis of correlation and response function corroborates the growth–climate relationships indicated by pointer year analysis. The dominant factor influencing tree-ring width in Scots pine is the temperature of the late winter/early spring period (February–March), and rainfall conditions in the May–July period (Figure 9). For BD, the dominant period of growth–climate relationship is in February and March, with positive correlation and response function values for both temperature and precipitation. Only positive values

occur for precipitation, additionally also in summer months (June–July), and in December of the year preceding growth. For temperature, statistically significant relationships are noted additionally in May and July, and in September of the year preceding growth (all values being negative). For YD, except for February and March, displaying correlation and response function values similar to BD, but lower, relationships occur for late spring and summer (May–July). These values are negative for temperature (May–July), but positive for precipitation (June–July). Negative values of statistical indices are noted for temperature in September of the year preceding growth, and positive values—for June. The dominant relationships for WB concern growth–precipitation (positive values of statistical indices except only for October of the preceding year) for February, May–July period, and June–July of the preceding year. Similar to the BD and YD populations, negative values of the correlation coefficient are observed for September of the year preceding growth, and positive values are observed for February (Figure 9).

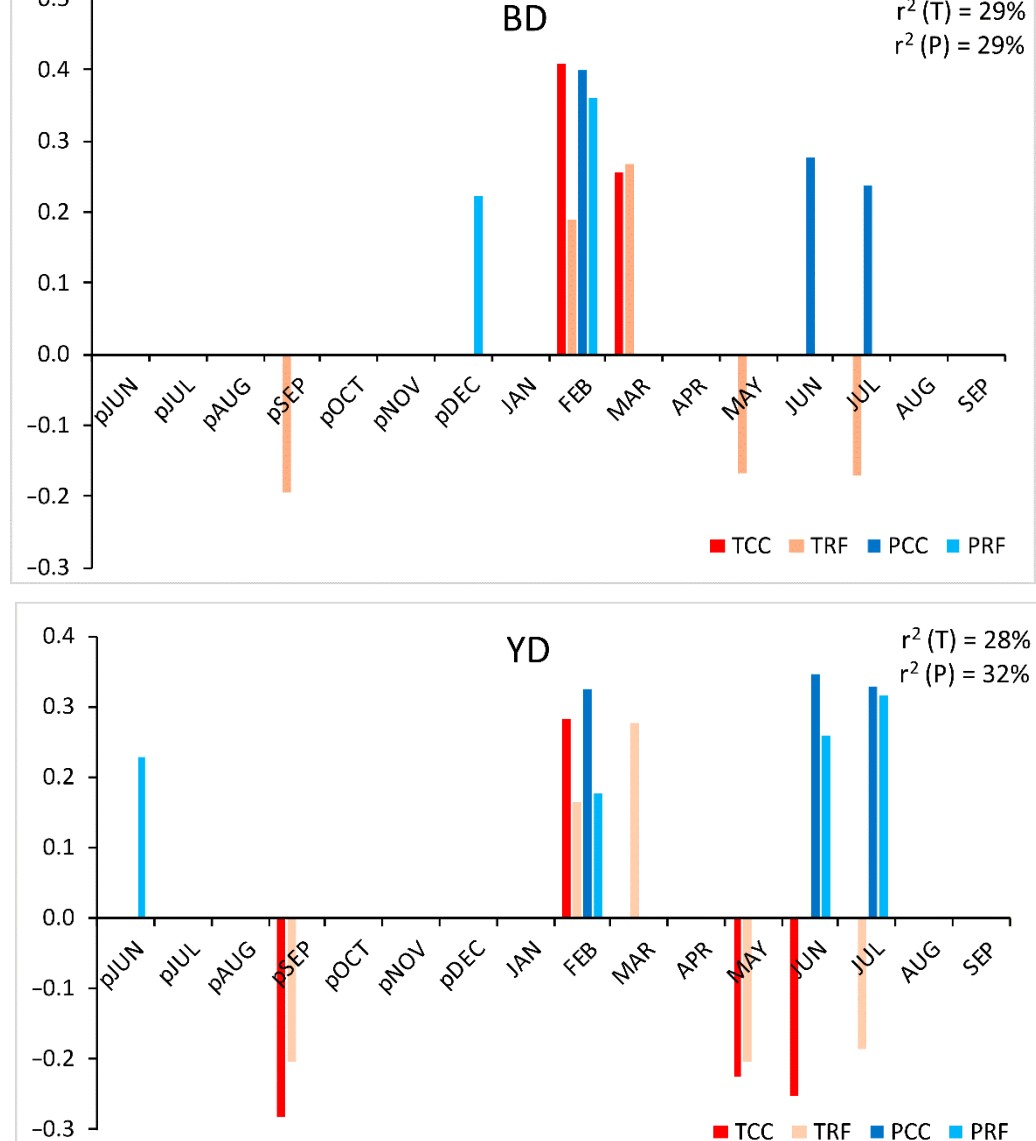

**Figure 9.** *Cont.*

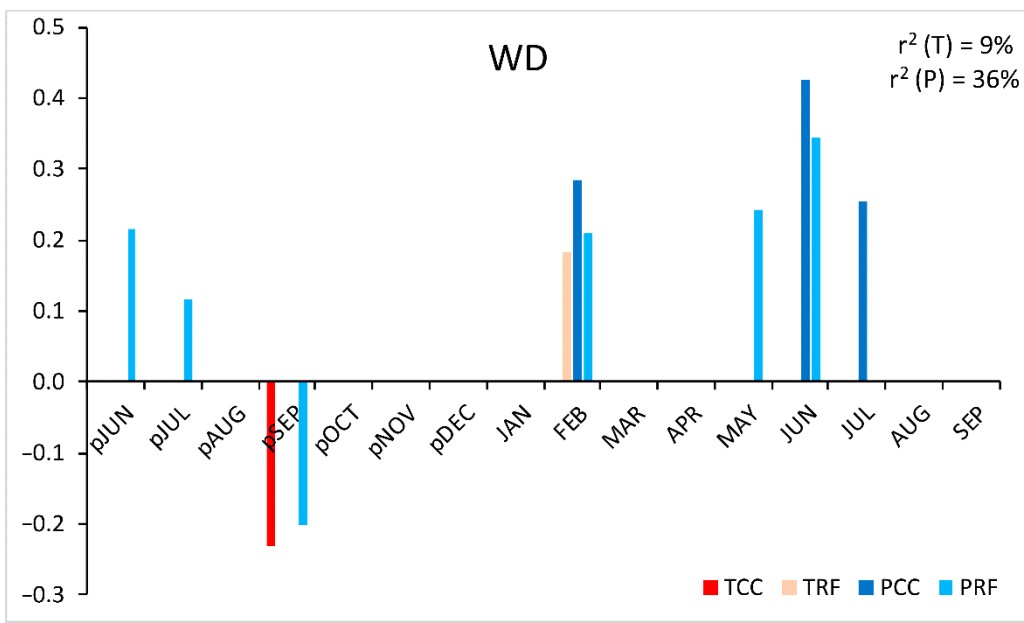

**Figure 9.** Results of correlation (CC) and response function (RF) analyses for Scots pine chronologies for temperature (T) and precipitation (P) in the period of 71 years (1948–2018). Bars denote significant values ($p \leq 0.05$); $p$, previous year; $r^2$, multiple regression determination coefficients; BD—brown dunes, YD—yellow dunes, WD—white dunes.

There are, however, considerable changes in the growth–climate response depending on the type/age of the dune. On brown dunes, temperature (T) and precipitation (P) have a similar importance for the tree-ring shaping process ($r^2$ for T and P = 29%). On yellow dunes, precipitation becomes the more important factor ($r^2$ for T = 28%, and for P = 32%). On the youngest dunes (white dunes), precipitation plays the dominant part, which is reflected in both $r^2$ values (for T = 9%, and for P = 36%) and in statistically significant correlation and regression values ($p = 0.0001$).

## 4. Discussion

Late winter and early spring temperature is the dominant climatic factor shaping tree-ring widths in Scots pine on the Southern and Eastern Baltic coasts [4,8,11,22,25,26,58,59]. The lack of strong frost in January, and especially in February, and an early and warm spring (March) result in good health of the trees, and the formation of broad tree-rings. The mechanisms underlying these relationships are not fully understood; most likely, with progressively longer days, and with hormonal changes taking place in Scots pines, the trees lose frost resistance and become vulnerable to low temperatures. An early onset of spring (temperature increase in March, lack of frost) extends the vegetation season and favors an acceleration of metabolic processes and cambial activity [6,27,60]. At numerous sites, Scots pines (especially those growing on sands) also display sensitivity to rainfall sums in summer (June–August) [20,26,59]. During summer droughts, sandy soils, which are characterized by high permeability and most frequently a deep groundwater table, suffer from water shortages in the tree root zone. Thus, trees growing on such substrata are prone to water stress [15].

Analyses of tree-rings in Scots pines growing on dunes or sandy areas on the Southern or Eastern Baltic coast are rather frequent. For instance, Pärn [61,62] examined the climate–growth relationship in pines of different ages, growing in various geomorphological settings on the dunes of Estonia. One of the findings of these studies was that tree age, but also microsite characteristics (elevation: western or eastern slopes of the dune and the position on the dune: foot or top of the dune) make an influence on growth reactions in Scots pine. Janecka et al. [26] studied the impact that the position of a tree

on a dune (on the ridge or at the foot) makes on ring-width variability, basal area increment, late wood blue intensity, and climate sensitivity. The microhabitat differences were significant only in some localities. On the other hand, winter–spring temperature and winter–spring/summer moisture availability had the dominant impact in all habitats. Furthermore, Mandre et al. [63,64] studied the influence that the height at which pines grow on a dune makes on the anatomical structure, morphological parameters, nutrient accumulation and biochemical characteristics of needles and radial growth of pines (study conducted on the coastal sand dunes in the southwestern Estonian dune field). The results of these analyses indicate water and nutrient shortages as the main factor differentiating tree-ring growth in Scots pines growing at various heights on a dune. Especially interdunal depressions, where organic matter accumulates, and soils are more nutrient- and water-rich, are a more favorable habitat for trees in comparison to dune slopes [65].

Vitas [66] studied pines growing on sandy soils about 200 m from the Baltic coast in northwestern Lithuania. With respect to substratum and distance from the coast, the location of the study site of Vitas [66] is similar to that of the WD site discussed in the present work. The trees studied by Vitas [66] are over 180 years old (at WD site—about 160 years). Vitas [66] reported a relationship between tree-ring width and air temperature in February and September, and weaker correlations with rainfall sums from the June–August period. As the two study sites are separated by a large distance, the pointer years determined by Vitas [66] poorly correspond to those determined for the BD, YD, and WD chronologies. A growth–climate analysis for Scots pines growing on the Southern Baltic coast is also presented by Cedro et al. [67]. In addition to the native pine species, Cedro et al. [67] also studied *Pinus banksiana* and *P. rigida*, planted in early 20th century for coastal dune stabilization. Additionally, in this case, the February–March air temperature is the dominant factor. Summer droughts are an additional growth-limiting factor [67].

Pointer years determined for pine chronologies from the Pomeranian Bay coastal dunes are consistent with pointer years determined for pines from northern Poland [4,59,67–69]. The dominant factor influencing tree-ring width is the February and March air temperature—e.g., the year 1940 is characterized by very narrow tree-rings over most of Poland [4,59,67,70,71]. This was due to a very cold winter: in comparison to a multi-year average, the temperature of January 1940 was about 9 °C lower, February—over 8 °C lower, and March—over 2.5 °C lower. Furthermore, the year 2006, not as cold as 1940, but with winter and spring considerably colder than average, saw the formation of very narrow tree-rings in this region. Not all pointer years, however, are linked to the late winter/early spring air temperature. Summer rainfall is an additional factor influencing cambial activity. Rainfall deficit in summer causes growth depressions (e.g., in 1973 or 1989), and higher than average rainfall, and lack of periods without rainfall cause the formation of wide tree-rings in pines growing in northern Poland (e.g., 1972 or 1997) [4,70,71].

Despite so many dendrochronological papers focusing on Scots pine from the Southern and Eastern Baltic coasts, studies on the influence of substrate fertility/soil type on tree-ring formation and growth–climate relationship are lacking. The soil profile structure (including ortstein horizon occurrence), sorption complex saturation, groundwater table depth, or soil capacity for water retention are highly important factors influencing tree health, and thus also cambial activity, and formation of tree-rings characterized by width varying on a year-to-year basis [72,73].

In extremely poor soil habitats developed on aeolian sands, devoid of mineral colloids, humus colloids make the strongest impact on the availability of nutrients for plants. Due to low biological activity and the resultant poor organic matter decomposition, even those substances are sparse. Soils developed on dunes are usually also strongly acidic, although Isserman [74] asserted that yellow dunes mostly show a neutral or basic soil reaction, gray dunes are moderately acidic, and brown dunes have a highly acidic soil reaction. Such properties, however, are not observed in the studied soils. This is because regardless of the degree of podzolization, the studied soils display a strongly acidic reaction, with slightly lower pH in older soils.

Habitat fertility is thus determined mostly by the upper accumulation horizon. An increase in organic matter accumulation is observed with soil development on dunes, reflected in both carbon quantity, and in thickness of the organic and humus horizons. At our study sites, these changes are especially well visible in the total Corg stock: the oldest soils contain six times more Corg than the youngest soils. Such disproportion in one of the key properties of soils formed on sands subsequently makes an impact on trees growing in these habitats. Some workers also point to the local topography exerting control on habitat fertility. This is because interdunal depressions are usually more fertile than the dunes [65,75,76].

**5. Conclusions**

For *P. sylvestris* (BD, YD, WD) growing on Świna Gate dunes on a similar substratum (aeolian sands), and in the same climatic conditions, soil richness is the main factor influencing growth dynamics. It shapes both tree growth rate (throughout their growth period) and growth–climate relationship. The oldest podzol that developed on brown dunes (BW, about 2500 years old), offers the best conditions for tree growth because of the quantity of accumulated organic matter, sorption complex saturation, and air–water properties. The growth–climate relationships observed here are typical for Scots pine growing on the Southern Baltic Sea coast (the main factor influencing the tree-ring width is late winter/early spring air temperature). The absence of strong frost, and early and warm spring make a positive influence on tree-ring width in the upcoming growing season. The less-well-developed podzol formed on the yellow dune (YD, 1600–1700 years of development) is characterized by a diminished organic matter quantity accumulated in the upper soil horizons, a weaker sorption complex, and a lower water retention capacity. This impacts both the growth rate (lower than in trees growing on brown dune), and the increase in the significance of rainfall in the summer period. On the white dune (WD), which is only about 400 years old and thus the youngest, pedogenetic processes thus far have only led to the transformation of regosol into a weakly developed podzol characterized by a shallow and poorly diverse soil profile, low organic matter quantity and a weak sorption complex. Such weakly transformed parent material is the poorest habitat for trees, which is reflected in the low growth rate (almost half the rate observed for BD), and in dissimilar growth–climate relationships. Despite the same precipitation sum and rainfall distribution through the year, the observed dominant relationship is the growth–rainfall relationship, which is uncommon for Scots pine. The reason for this is the host rock: a highly permeable sand unaltered by soil-forming processes at depths as shallow as 50–60 cm.

As no similar studies are available, additional analyses and further studies are required on growth–climate relationship in trees stabilizing dunes of different ages, and offering the main protection for marine coast. In today's period of rising sea levels and increasingly intensive storms threatening dune barriers, it is essential to gain an understanding of all ecosystem interrelationships. Plants growing on dune barriers (forest in particular) represent the best natural protection against storm floods and coastal erosion. In recent decades, characterized by climate changes and a progressively more intense coastal zone economy, preventing coast erosion generates increasingly higher costs.

**Author Contributions:** A.C., B.C., conceptualization, field collection; A.C., B.C., M.P., methodology, data analyses, writing—draft preparation, review and editing. All authors have read and agreed to the published version of the manuscript.

**Funding:** This research received no external funding.

**Informed Consent Statement:** Not applicable.

**Data Availability Statement:** Data available in a publicly accessible repository. The data presented in this study are available in RepOD at https://doi.org/10.18150/1O3EBD, accessed on 3 February 2022.

**Conflicts of Interest:** The authors declare no conflict of interest.

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
