# Peer review of "Differences in Growth–Climate Relationships among Scots Pines Growing on Various Dune Generations on the Southern Baltic Coast"

_forests, doi:10.3390/f13030470_

Round 1

Reviewer 1 Report

Review of the submitted manuscript entitled: Differences in growth-climate relationships among Scots pines growing on various dune generations on the Southern Baltic coast

This manuscript presents the results of dendrochronological studies conducted on three types of coastal dunes with different soil properties. The research showed that the fertility of the dunes had a strong influence on the chronology of the Scots pine tree-rings. The correlation between the tree-rings widths, meteorological conditions, and pointer years was similar to those documented in other studies in Poland (e.g. Klisz et al. 2019, Misi et al. 2019). Research results suggest that when interpreting the results of dendrochronology studies, more attention should be paid to edaphic factors. The results are also potentially useful for forest management. Knowing about the differences in the incremental responses of trees on different soil types in the dunes can help plan tree cultivation. On such permeable, sandy soils, especially in the higher positions of the dunes, there is an exceptionally high risk of water scarcity, which will increase with the following climate warming.

The aim of the research is precise and focused on getting to know them so far poorly researched research thread of potentially valuable for forestry. The materials and methods are standard, known from many other dendrochronological studies. Technically correct, but more detailed sampling information for tree-rings is missing. Please add the information about the topographic position of study sites and the selection of trees for sampling. It is essential because the slopes, tops of the dunes and their depressions differ significantly in soil and meteorological conditions (e.g. Sewerniak 2016, Sewerniak and Puchałka 2020).

I have no objection to the description of the results. Figures and tables are legible and informative. However, authors should use a period as the decimal separator in figures. In general, the Discussion is interesting, but some paragraphs are not supported by a quotation from the literature and contain content not directly related to this research results. The excerpt between lines 376 and 404 does not include any references to the literature. It also does not apply to the authors' studies described in this manuscript. The Conclusions sound more like summaries of the most important results than conclusions. Here is a place for more speculative content or indicating gaps in knowledge and direction for further research.

Specific comments:

L.96: soil-froming->soil-forming

L.332: There is no reference to literature at the end of this paragraph. In Poland, habitat studies on dunes were conducted by Sewerniak and Puchałka (2020).

L338-340: Similar pointer years and correlations with climatic data have been also found in other dendrochronological studies on pine in N Poland (see Misi et al 2019; Klisz et al. 2019).

L.357: There is no citation at the end of the paragraph due to a lack of similar research. However, I think that Sewerniak (2016) can be quoted here. This article does not describe a dendrochronological study but is appropriate to be cited here.

L.364: There is no reference to own results at the end of the sentence.

L.369: There is no reference to the literature.

L.381: It is unclear what the region means - the Baltic coast, northern Poland, other, central Europe, other? There is also no reference to literature.

L.376-396: Paragraph without support from literature citations and direct link to research results. There are no citations of literature, tables or figures.

L.397-404: This excerpt with minor changes may be replaced to Conclusions.

All Latin species names should be written in italics. The comment covers the entire manuscript.

References

Klisz, M., Puchałka, R., Wilczyński, S., Kantorowicz, W., Jabłoński, T., & Kowalczyk, J. (2019). The effect of insect defoliations and seed production on the dynamics of radial growth synchrony among Scots pine Pinus sylvestris L. provenances. Forests, 10(10), 934. doi: 10.3390/f10100934

Misi, D., Puchałka, R., Pearson, C., Robertson, I., & Koprowski, M. (2019). Differences in the climate-growth relationship of Scots Pine: a case study from Poland and Hungary. Forests, 10(3), 243. doi: 10.3390/f10030243

Sewerniak, P., & Puchałka, R. (2020). Topographically induced variation of microclimatic and soil conditions drives ground vegetation diversity in managed Scots pine stands on inland dunes. Agricultural and Forest Meteorology, 291, 108054. doi: 10.1016/j.agrformet.2020.108054

Sewerniak, P. (2016). Impact of land relief on site index and growth parameters of Scots pine stands on inland dunes in the Torun Basin. Sylwan, 160(8), 647-655.

Kind regards

Author Response

Szczecin, 10.03.2022

Reviewer 1

Manuscript ID: forests-1605456

Title: Differences in growth-climate relationships among Scots pines growing on various dune generations on the Southern Baltic coast

Authors: Anna Cedro, Bernard Cedro, Marek Podlasiński

Received: 4 February 2022

We sincerely thank for the insightful analysis of our text, all the comments, and suggestions. We did our best to take into account all the remarks.  We hope that the revised manuscript now meets the requirements of the editors and reviewers, and thus is suitable for publication.

In part 2.2 Tree-ring Data, information about the topographic position of the study sites and the selection of trees for sampling was added;

In figures and tables have replaced commas with dots;

Conclusions were corrected according to the reviewer’s suggestion.

Line 96 - has been changed, according to the reviewer’s suggestion;

Line 332 - has been changed, according to the reviewer’s suggestion (the reference has been added);

Line 338-340 - has been changed, according to the reviewer’s suggestion (the reference has been added);

Line 357 -  has been changed, according to the reviewer’s suggestion (the reference has been added);

Line 364 - has been changed, according to the reviewer’s suggestion (the reference has been added);

Line 369 - has been changed, according to the reviewer’s suggestion (the reference has been added);

Line 381 - has been changed, according to the reviewer’s suggestion (the region was specified  and here we summarize the results of our research, therefore there is no citation);

Line 376-396 - here we summarize the results of our research, therefore there is no citation, has been revised to make it easier to identify our studies;

Line 397-404 - has been changed, according to the reviewer’s suggestion;

All Latin species names were written in italics.

Thank you for pointing to additional references.

Thank you very much for all your comments.

Anna Cedro, Bernard Cedro, Marek Podlasiński

Reviewer 2 Report

The manuscript tends to explore the relationship between the physical features of the tree growth rings of the most frequently applied tree species i.e. the Scot pine and meteorological parameters that include rainfall and temperature along with the pedological features of the scot pine stands in the northern part of Poland.

I have found the work very interesting, containing valuable data, which the authors have obtained from extensive fieldwork and rigorous lab work.

In the abstract and material methods sections, the authors have talked about the relationship among tree growth rings, meteorological factors, and soil variables. Based on what they wrote in those sections, the reader of their manuscript, including me, would be very eager to read more about their findings regarding those relationships and to be more specific regarding the regression models. However, I could not find what I expected once I got to the results section of the manuscript.

So, I strongly recommend to open up a subsection under the title of "Regression Analyses", in the Results Section of their manuscript to report the related results regarding the regression models, by which other scientists and researchers could apply their developed regression models for their own uses. 

Author Response

Szczecin, 10.03.2022

Reviewer 2

Manuscript ID: forests-1605456

Title: Differences in growth-climate relationships among Scots pines growing on various dune generations on the Southern Baltic coast

Authors: Anna Cedro, Bernard Cedro, Marek Podlasiński

Received: 4 February 2022

We sincerely thank for the insightful analysis of our text, all the comments, and suggestions. We did our best to take into account all the remarks.  We hope that the revised manuscript now meets the requirements of the editors and reviewers, and thus is suitable for publication.

In the analyzes, meteorological data – tree-ring width, a simple (linear) regression analysis was used, and due to the lack of space and standard methods used, we do not describe them in more detail, but refer to other publications where they are used and described.

But the description of the results obtained in the correlation and response function analysis has been significantly expanded according to the reviewer’s suggestion;

The results of this analysis are described in Results, subsection 3.3.2. Correlation and response function analysis.

In the analyzes of soil properties-tree-ring width, we could not apply the regression analysis, because we only have data for three sampling sites (single sampling) and it cannot be compared in the annual resolution data (tree-ring width).

Thank you very much for all your comments.

Anna Cedro, Bernard Cedro, Marek Podlasiński

Reviewer 3 Report

The manuscript about the growth-climate relationships of Scots pine on different dune generations compares the response to precipitation and temperature of Scots pine, using tree-ring width data and comparing the response on three dune generation. Although the manuscript gives some new information about the response of the Scots pine growth, the provided information in the manuscript should rearrange to better show methods used and the results obtained. The manuscript has lot of information about the soils but there is no statistical analysis to compare three dune types (with only 13 sampling plots it would also be hard to do such tests). So, the information about the soils should be reduced as only descriptive part. As the authors use both tree-ring width data and pointer year data, the results should clearly show the common and different aspect regarding the response to climatic data.

Introduction

Lines 42-43 – species Latin names should be written in italic

Methods

Please add software names and version used for the data analysis.

Lines 118 – why some trees have two radii and some only one. Please add some explanation.

Lines 146-160 – please move soil analysis to separate paragraph as title “Tree-ring Data” is misleading in this case. Also, the part on data analysis could be written as a separate paragraph.

Line 161 – please add some figure showing climatic data as they are significant part of this manuscript, especially in the context of pointer year analysis.

Figure 1 – OSL datums numbers for some plots are hard to see and, they are missing for some plots. Maybe move datums to table, describing sampling-plots

Results

Line 254 – please consider renaming chapter “Dendroclimatology” to something more meaningful.

Line 274 – the results parts mention “correlation and response function” but methods part “correlation and regression” analysis. Are those the same analysis?

Lines 280-281 – authors mention that precipitation is more significant factor for yellow dunes, but the difference is only 3 percentage points and there is no test (for example, randomisation test) to prove that difference is significant.

Figure 4 – Please use only one form for data representation – bars or lines as they both show the same information

Figure 8 – there is only one r2 value in each subplot for T or P. Is this mean value? It is not possible to understand from the plot or from the methods part.

Table 3 – Total number of samples according to the table is 61, methods part (Line 118) states that 68 tree were sampled. Please add some explanation text to manuscript why 7 tree were excluded

Author Response

Szczecin, 10.03.2022

Reviewer 3

Manuscript ID: forests-1605456

Title: Differences in growth-climate relationships among Scots pines growing on various dune generations on the Southern Baltic coast

Authors: Anna Cedro, Bernard Cedro, Marek Podlasiński

Received: 4 February 2022

We sincerely thank for the insightful analysis of our text, all the comments, and suggestions. We did our best to take into account all the remarks.  We hope that the revised manuscript now meets the requirements of the editors and reviewers, and thus is suitable for publication.

Line 42-43 – Latin names were written in italics, according to the reviewer’s suggestion;

Methods – we added version and software names, according to the reviewer’s suggestion;

Line 118 – we added explanation about the method of sampling and using samples to build a chronology, according to the reviewer’s suggestion;

Lines 146-160 – we separate paragraph 2.3. Sand and Soil Analysis, and 3.1. Aeolian sands and soils (in Results) according to the reviewer’s suggestion;

Line 161 – we added figure showing climatic data, according to the reviewer’s suggestion;

Figure 1 – OSL dates have been enlarged to increase their readability, according to the reviewer’s suggestion;

Results

Line 254 – the name of the "Dendroclimatology" section was changed to "Dendroclimatology analysis" and the section was divided into sub-sections: "Pointer Years Analysis" and "Correlation and response function analysis", according to the reviewer’s suggestion;

Line 274 –  this term will be renamed “correlation and response function” throughout the paper, thank you for pointing out a mistake;

Lines 280-281 – the word significant was used incorrectly here, it was changed to important, the correlation significance test was performed and indeed only for white dunes (WD) the differences are statistically found, thank you for pointing out the error;

significance test, the difference between the two structure indices, N=71

BD: r2 (T) =29%, r2 (P)= 29%       p=0.5

YD: r2 (T) =28%, r2 (P) = 32%      p=0.3015

WD: r2 (T) =9%, r2 (P) = 36%      p=0.0001

Figure 4 – has been changed, according to the reviewer’s suggestion;

Figure 8 – the analysis was performed for the period 1948-2018 (71 years) and the r2 value refers to this period for a given meteorological parameter; the caption under the figure has been supplemented, according to the reviewer’s suggestion;

Table 3 – an explanation of the difference between the number of samples taken and the samples used to build the chronology was added in the section Material and Methods, according to the reviewer’s suggestion.

Thank you very much for all your comments.

Anna Cedro, Bernard Cedro, Marek Podlasiński

Reviewer 4 Report

The paper analyses relation between climate factors and radial growth rate for Scots pine stems grown on podzols formed on three different types of dunes. The research has a lot of data related to pedology, dendrochronology and dendroclimatology. I would really like to brag the paper and no doubt it is of a very high quality. There are just two short questions that can be understood as suggestions - you mentioned 10 frost rings when you conducted dendrochronolohgical analysis. I am not sure if these rings are false or not? What actually define this term? The another recommendation is connected with the final part in the Discussion - you described the importance of Scots pine in coastal areas because they prevent from rising sea levels and strong storms. Is it possible to plant some other species for the same purpose? Have you been thinking about introducing some special genotypes of Scots pine based on genetics-physiological basis that would be adequate for preventing from coastal erosion? 

All in all, I am very satisfied with the paper.

Author Response

Szczecin, 10.03.2022

Reviewer 4

Manuscript ID: forests-1605456

Title: Differences in growth-climate relationships among Scots pines growing on various dune generations on the Southern Baltic coast

Authors: Anna Cedro, Bernard Cedro, Marek Podlasiński

Received: 4 February 2022

We sincerely thank for the insightful analysis of our text, all the comments, and suggestions. We did our best to take into account all the remarks.  We hope that the revised manuscript now meets the requirements of the editors and reviewers, and thus is suitable for publication.

The emergence of frost rings (deformations) is associated with extreme weather conditions, mainly with the advection of cold air masses and temperature drop below 00C during the growing period (particularly in late spring and summer, and also in early autumn). The temperature drop below the freezing point may occur during the night, whereas the temperature during the day is above zero. The low temperatures induce the formation of ice crystals in strongly hydrated xylem tissues, which results in tissue dehydration and deformation of vessels, tracheids and cells (Fig. 1 and 2).

An explanation has been added in the text, according to the reviewer’s suggestion.

Fig. 1. Pseudotsuga menziesii, Frost ring in early wood dated to 1913. Visible damages of cells and radial deformations (Cedro 2004).

Fig. 2. Frost rings in consecutive years in a cross-section at 100x magnification, tree no. TA16 (Sorbus torminalis L.) (Cedro 2016).

The use of other pine species to stabilize coastal dunes is described in the study by Cedro et al. 2013 (in Discussion). Three pine species were analyzed there: Pinus sylvestris, P. banksiana and P. rigida. All the trees grew in the same habitat and were of the same age. The highest tree-ring width were found for P. rigida, and the lowest for P. sylvestris. However, these differences were slight. Besides, coastal areas are often covered by various forms of protection (e.g. Natura 2000) and not native tree species cannot be introduced there. That is why the native Scots pine is used for planting on newly formed dunes or for restoration in coastal dunes.

Thank you very much for all your comments.

Anna Cedro, Bernard Cedro, Marek Podlasiński

Round 2

Reviewer 1 Report

The authors responded to most of my comments. However still, some technical corrections are needed.

Dendroclimatology Analysis -> Dendroclimatological Analyzes

The full species name of Pinus sylvestris should only be used by the authors when it is first mentioned when it is mentioned in the main text. 
Then, each subsequent time the short name P. sylvestris should be used.

All Latin names for species should be written in italics.

L.434-455: This section still looks more like a summary of the main results and conclusions than the Discussion. The authors should support this paragraph with literature references or move this fragment to Conclusions. 

Kind regards

Author Response

Szczecin, 14.03.2022

Cover letter

We are very grateful to Editor and Reviewers for very relevant suggestions, which enables to improve the quality of our paper. We have tried to take account suggestions of all the reviewers in text.

The changes made in the “Track Changes” function.  

Dendroclimatology Analysis -> Dendroclimatological Analysis -  corrected by suggestions of reviewer

The full species name of Pinus sylvestris should only be used by the authors when it is first mentioned when it is mentioned in the main text. 
Then, each subsequent time the short name P. sylvestris should be used -  corrected by suggestions of reviewer

All Latin names for species should be written in italics -  corrected by suggestions of reviewer

L.434-455: This section still looks more like a summary of the main results and conclusions than the Discussion. The authors should support this paragraph with literature references or move this fragment to Conclusions -  corrected by suggestions of reviewer

(x) English language and style are fine/minor spell check required - The English language was checked for style and correctness by a native speaker, changes were made, corrected by suggestions of reviewer

We hope that the changes that were made in paper will improve quality and an article to be published in MDPI Forests.

Best Regards,

Anna Cedro

Bernard Cedro

Marek Podlasiński

Reviewer 2 Report

Congrad. for your paper.

Author Response

(The authors gave the same response as above.)

Reviewer 3 Report

Thank you to authors for improving the manuscript and answering to all suggestions raised by reviewer.

Author Response

(The authors gave the same response as above.)
